# Inflammaging: The Next Challenge—Exploring the Role of Gut Microbiota, Environmental Factors, and Sex Differences

**DOI:** 10.3390/biomedicines12081716

**Published:** 2024-08-01

**Authors:** Mario Caldarelli, Pierluigi Rio, Andrea Marrone, Vincenzo Giambra, Antonio Gasbarrini, Giovanni Gambassi, Rossella Cianci

**Affiliations:** 1Department of Translational Medicine and Surgery, Catholic University of Sacred Heart, 00168 Rome, Italy; mario.caldarelli01@icatt.it (M.C.); pierluigi.rio01@icatt.it (P.R.); andrea.marrone01@icatt.it (A.M.); antonio.gasbarrini@unicatt.it (A.G.); giovanni.gambassi@unicatt.it (G.G.); 2Fondazione Policlinico Universitario A. Gemelli, Istituto di Ricerca e Cura a Carattere Scientifico (IRCCS), 00168 Rome, Italy; 3Institute for Stem Cell Biology, Regenerative Medicine and Innovative Therapies (ISBReMIT), Fondazione IRCCS “Casa Sollievo della Sofferenza”, 71013 San Giovanni Rotondo, Italy; v.giambra@operapadrepio.it

**Keywords:** aging, inflammation, immunosenescence, gut microbiota, diet, sex differences

## Abstract

The term ‘inflammaging’ has been coined to describe the chronic state of inflammation derived from ongoing cycles of tissue damage and the subsequent immune responses. This inflammatory status contributes to the decline of organs and physiological functions, accelerates the aging process, and increases the risk of age-related illnesses and death. During aging, the gut microbiota (GM) undergoes significant changes, including a decreased diversity of species, a decline in beneficial bacteria, and a rise in proinflammatory ones, resulting in persistent low-grade inflammation. Moreover, environmental factors, such as diet and medications, contribute to age-related changes in GM and immune function, preventing or promoting inflammaging. This narrative review aims to clarify the underlying mechanisms of inflammaging and to specifically investigate the influence of GM and several environmental factors on these mechanisms, while also exploring potential differences related to sex. Moreover, lifestyle and pharmacological interventions will be suggested to promote healthy aging.

## 1. Introduction

From a biological perspective, aging is an intricate process encompassing both environmental and genetic factors. Despite an increase in human life expectancy, aging is characterized by an impairment of body functions and a consequent development of diseases, which compromise human well-being. Among the aging-related alterations impacting human health, immunological imbalance, inflammation, DNA mutations, and mitochondrial failure play a crucial role [1].

As observed by Singh et al., aging results from the progression of disequilibrium between damage accumulation and the progressive inefficiency of the repair mechanisms, leading to disease, frailty, and ultimately death [2]. The immune system, which represents the first line in the resilience mechanisms, goes through an aging-related hyperactivation with increased levels of inflammatory mediators in blood and tissues, defined as ‘inflammaging’ [2]. Moreover, with aging, an impairment in both innate and adaptive immune responses against pathogens is commonly observed, thus increasing morbidity and mortality. This complex and dynamic condition is commonly defined as ‘immunosenescence’ and encompasses several mechanisms, such as persistent low-grade inflammation, a higher number of memory T cells, and a reduced ability to organize immune responses to antigens [3]. For a long time, immunosenescence has been considered an unfavorable maladaptive phenomenon, being associated with lower humoral and cellular responses against infections and vaccines. Interestingly, in recent years, several researchers have revised this concept, suggesting that age-related immune remodeling is essential to respond to detrimental stimuli, as observed for instance in centenarians. In the age-related proinflammatory context, the hyperactivation of the immune system may also provide anti-inflammatory responses with a positive impact on longevity [4].

The existence of sex differences in the trajectories of aging has been extensively reported. Despite a greater occurrence of diseases, women have a longer life expectancy compared to men, probably due to sex-specific biological clocks [5].

The gut microbiota (GM) significantly influences the aging process, marked by notable shifts in its composition, including an increase in proinflammatory species and a reduction in beneficial microbes. Both internal and external factors, such as diet, medications, and chronic illnesses, contribute to changes in the GM, which in turn influence age-related inflammatory pathways. Since the GM takes part in the complex crosstalk with the immune system, its role as a determinant of healthy aging as well as inflammaging has been evaluated [6].

This narrative review aims to clarify the immunobiology of inflammaging and the impact of aging on the GM, with a focus on the potential role of sex differences in these mechanisms. Understanding how immunity, the GM, and environmental factors act in elderly individuals may allow the design of tailored therapeutic interventions aimed at promoting healthy aging. The literature review was conducted in June 2024 using scientific databases, including PubMed, Google Scholar, MEDLINE, and Embase. 

The following specific keywords were employed to refine the search: “inflammaging”, “sex differences”, “gut microbiome”, and “immunosenescence”. We considered original and review articles, meta-analyses, and systematic reviews focusing on the GM, inflammaging, and sex differences, written in English in the last ten years, and published only in peer-reviewed journals. The quality of the included studies was based on study design, methodology, sample size, and relevance to the research questions. Data from studies with a robust methodology were analyzed to achieve an extensive overview of the current understanding of the topic.

## 2. Immunosenescence and Inflammaging

Immunosenescence involves several biological changes, such as the involution of the thymus, the dysfunction of hematopoietic stem cells, alterations in T and B cell subpopulations, and the development of a senescence-associated secretory phenotype (SASP) by senescent immune cells [7].

The secretion of proinflammatory mediators, such as interleukin (IL)-1, IL-6, IL-8, IL-18, tumor necrosis factor (TNF), and interferon-γ (IFN-γ), is one of the main hallmarks of immunosenescence [7]. A recent Brazilian study on immunosenescence biomarker profiles in older adults found that increased levels of the proinflammatory C-XC motif chemokine ligand 10 (CXCL10), IL-2, IL-4, IL-6, IL-17, TNF-α, and C-C motif chemokine ligand 11 (CCL11) were positively associated with the progression of age, whereas the anti-inflammatory IL-1 receptor antagonist (IL-1Ra) had a negative association [8]. Tylutka et al. considered IL-6 to be an important marker of peripheral inflammation in patients with age-related diseases [9]. The influence of sex on the blood concentration of inflammatory mediators has been detected, since adult men exhibit higher levels of IL-1β, IL-6, and TNF-α, compared to women. On the contrary, anti-inflammatory stimuli, such as greater levels of IL-10, transforming growth factor beta (TGF-β), and IL-1Ra, are predominant in women [10]. The interplay of genetic, epigenetic, and hormonal factors is responsible for the existence of sex differences in immune and inflammatory mechanisms [11]. Women experience more frequently autoimmune disorders throughout life, whereas men are more prone to infectious and chronic inflammatory diseases. With aging, particularly after menopause, the differences in immune function tend to decrease, leading to a similar incidence of autoimmune and chronic inflammatory disorders in aged individuals independent of gender [12].

In response to sex hormones, women have an increased CD4/CD8 T lymphocyte ratio compared to men, who exhibit a predominance of CD8 T lymphocytes and regulatory T cells (Tregs) which have immunosuppressive roles [13]. It has been observed that estradiol promotes the activation of CD4^+^ T cells, whereas progesterone has the opposite effect. Moreover, testosterone inhibits the differentiation of T helper 1 (Th1) cells and upregulates the expression of IL-10, which encourages a Th2-type immune response [11]. The protective function of the immune system physiologically declines in older individuals, in parallel with the amplified systemic inflammation [13].

The age-related thymus degeneration leads to a reduction in naïve T cells, an increase in memory T cells, and a decline in T cell output. At the same time, changes in T cell profiles, including a reduced T cell receptor (TCR) diversity and a lower antigen recognition, have been observed [7]. The age-associated thymus decline may also break the immune tolerance to self, thus leading to the onset of autoimmune disorders, which indeed occur more frequently with aging [14]. Furthermore, both T and B cell responses decline quantitatively as well as qualitatively. In this context, conditions typically associated with a reduced life expectancy are a significantly decreased number of B cells, the reversal of the CD4/CD8 T cell ratio, and lower responses against mitogenic stimuli [15]. A study evaluating the markers of severity of COVID-19 in older populations found that increased terminally differentiated CD8^+^ T cells, decreased CD8^+^ T stem cell-like memory cells, and higher levels of proinflammatory cytokines and chemokines significantly correlate with the disease severity [16]. Other hallmarks of aging could be an increased number of cytotoxic natural killer (NK) cells and the inefficacy of phagocytosis and oxidative burst by neutrophils [17].

Aging is characterized by metabolic changes, such as increased glycolysis, mitochondrial dysfunction, and a higher release of reactive oxygen species (ROS) [7]. A decline in fatty acid oxidation (FAO), particularly in macrophages, monocytes, and microglia, occurs with aging.

As described by Kruglov et al., chronically stimulated pathogen- and damage-associated molecular patterns (PAMPs and DAMPs) activate inflammatory pathways, including the NOD-like receptor family pyrin domain containing 3 (NLRP3) inflammasome. This promotes macrophage proinflammatory polarization, characterized by enhanced glycolysis, downregulation of peroxisome proliferator-activated receptor-α (PPAR-α) and FAO, increased triglyceride synthesis, and reduced 5′ AMP-activated protein kinase (AMPK) signaling, which impairs the monocyte-to-macrophage differentiation and increases the ROS production [18]. Sexual dimorphism in ROS balance exists since men display an increased ROS production at the mitochondrial electron transport chain and weaker antioxidant mechanisms, as compared to women. This results in a higher risk of oxidative damage in macromolecules, such as DNA, proteins, and lipids, which causes an imbalance in cellular functions. The increased oxidation and inflammation could explain the shorter lifespan of men. However, after menopause, depletion of estrogens, which exert antioxidant functions, increases oxidative damage in women, causing less pronounced differences between the genders [19].

Salminen et al. has recently evaluated the potential role of tissue fibroblasts as immune regulators in the aging process [20]. Fibroblasts are versatile stromal cells that can exhibit an immune-related gene expression and can interact with immune cells, acting in a pro- or anti-inflammatory way depending on the context. It has been observed that, in aged tissues, fibroblasts express genes encoding for chemokines and complement components, and have less fibrogenic properties [20].

Genome instability is considered a shared feature of both aging and cancer. DNA damage and the consequent activation of damage response systems could drive human aging through different mechanisms, such as a direct impact on DNA replication and transcription, the induction of cell death, or the development of DNA mutations [21]. In this context, telomere shortening is a known hallmark of senescence, associated with the occurrence of aging-related diseases [22]. Telomeres play an important role in protecting chromosomes from the activation of DNA damage responses; however, they undergo a progressive reduction during cell division, which may result in genomic instability, senescence, and ultimately cell death [23]. Telomere length in leukocytes is highly variable, and women generally have longer telomeres, compared to men, at any age. Hence, Lansdorp hypothesized that higher levels of embryonic telomerase, encoded by *DKC1* (dyskerin pseudouridine synthase 1) on the X chromosome, in women could explain this sex dimorphism [24]. As observed in a Mendelian randomization study conducted by Wang et al., telomere shortening may increase the risk of a reduction in T cell percentage and impair the B cell differentiation (e.g., increased naïve–mature B cells and reduced memory B cells), leading to immunodeficiency that may explain a higher cancer predisposition with aging [25]. Moreover, various immune phenotypes of senescent T cells have been associated with malignant tumors, such as the downregulation of CD27 and CD28, and the upregulation of CD57, cytotoxic T-lymphocyte-associated protein 4 (CTLA-4) and killer cell lectin-like receptor subfamily G (KLRG-1) [25]. Additionally, age-related changes in the extracellular matrix may promote the development of cancer, as well as tumor progression and metastasis. In this context, the most important mechanisms are stromal cell disorganization, increased SASP of senescent cells, infiltration of immunosuppressive cells (e.g., myeloid-derived suppressor cells, MDSC, or Tregs), and loss of cytotoxic activity [26].

With aging, the emergence of maladaptive T and B cells (e.g., CD4^+^CD28^−^ and CD8^+^ Granzyme K (GzmK)^+^ T cells, and age-associated B cells) has also been observed in atherosclerosis. In addition, the relationship between adaptive immunity and atherosclerosis is characterized by reduced plasma levels of antibodies targeting oxidation-specific epitopes (OSEs) and increased BAFF (B cell activating factor), sustaining pro-atherogenic B cells, as illustrated in Figure 1 [27].

In recent years, the understanding of immunosenescence and inflammaging has significantly evolved. Although proinflammatory factors and inflammaging were once believed to be the primary causes of most age-related diseases, current insights reveal that age-related inflammation is more closely linked to longevity than to any other factors [28].

It has been suggested that immunosenescence and inflammation represent the two facets of aging. From an evolutionary perspective, immunosenescence is considered as an optimization of the aging body’s resources, even though it may eventually result in diseases and death [29].

Another possibility is that immunosenescence refers to the gradual decline of adaptive immunity over time, while inflammaging represents the activation of innate immunity [4].

Inflammaging and immunosenescence progress concurrently: the increased release of inflammatory mediators due to inflammaging inhibits the adaptive immune system and promotes immunosenescence, while the weakened response of the adaptive immune system enhances the stimulation of the innate immune system and promotes the inflammaging [28]. One of the latest hypotheses regarding the age-related changes in the immune system suggests that these modifications are linked to the immune system’s response to chronic stress. This new perspective considers these changes as a continuous process of adaptation to stress, which can have either beneficial or detrimental effects, depending on both genetic and environmental factors, leading to either healthy longevity or pathological aging characterized by age-related diseases [30].

As potential markers of immunosenescence, it has been proposed that T cells that exhibit a CD27^−^CD28^−^, CD57^+^ killer cell lectin-like receptor G1 (KLRG-1)^+^ or C-C chemokine receptor 7 (CCR7)^−^CD45RA^+^ phenotype may function as potential markers of immunosenescence. These cells may also express T cell immunoglobulin and mucin domain-containing 3 (Tim-3), T cell immunoreceptor with Ig and ITIM domains (TIGIT), immunoglobulin-like transcript 2 (ILT2/CD85j), or other NK-like receptors [31].

Moreover, terminal effector memory T cells (TEMRA), characterized by reduced proliferation and high clonal expansion, are considered hallmarks of immunosenescence [31]. Finally, senescent T cells exhibit positive staining for senescence-associated β-galactosidase (saβ-gal), upregulate p53, p21, and p16, downregulate cyclin-dependent kinase (Cdk) 2, Cdk6, and cyclin D3, and display a senescence-associated secretory phenotype (SASP) [32].

A mention should be made of the vaccine immune response in the elderly population, which is a matter of debate for modern research. It is known that older adults are more susceptible to infections and develop weaker responses to vaccines due to the senescence of the immune system [33]. Influenza, pneumonia, or infections by novel pathogens (e.g., severe acute respiratory syndrome coronavirus 2) have proved to be highly severe in the older population, and are often causes of death [28].

With aging, both innate and adaptive immune responses lose efficacy, thus contributing to lower vaccine immunogenicity. Among the underlying mechanisms, Allen et al. described the reduced function of macrophages, monocytes, neutrophils, and NK cells, impairment in T cell activation by dendritic cells, and changes in adaptive immunity, such as a reduced output of naïve cells, phenotypic changes in T and B cells, lower diversity, and impaired differentiation [34].

The mechanisms of immunosenescence and inflammaging mentioned above are summarized in Table 1.

## 3. Gut Microbiota and Inflammaging

The GM consists of a broad spectrum of microorganisms, including bacteria, viruses, protozoa, fungi, and archaea. In general, 99% of the species are examples of Firmicutes, Bacteroidetes, Proteobacteria, Actinobacteria, Fusobacteria, and Verrucomicrobia. These microorganisms can impact human health in both positive and negative ways [35].

Maintaining a healthy and stable GM is crucial, as it should be able to recover and return to its normal state after disruptions caused by several factors. If the GM cannot withstand these disturbances, irreversible changes can occur, resulting in a potentially harmful condition called dysbiosis [36]. Dysbiosis is marked by changes in the GM, such as an increase in proinflammatory species and a decrease in anti-inflammatory ones [37].

Aging can promote and progress the onset of inflammaging in the gut, which subsequently exacerbates the pathological changes through various mechanisms. These mechanisms include damaging the intestinal barrier, dysbiosis, and causing immunosenescence [38].

During aging, the GM suffers multiple changes, such as a decrease in the diversity of species, a rise in the prevalence of Proteobacteria, and a decline in the presence of probiotic bacteria, like *Bifidobacterium* [39].

The rise in proinflammatory bacteria can suppress the growth of beneficial bacteria, resulting in persistent low-level inflammation and heightening the risk of several age-related diseases [40]. Biagi et al. have demonstrated that elderly populations can have a reduced amount of *Bifidobacteria* and certain anti-inflammatory Firmicutes, including members of Clostridium cluster IV (such as *Ruminococcus obeum*, *Roseburia intestinalis*, *Eubacterium ventriosum*, *Eubacterium rectale*, and *Eubacterium hallii*) and Clostridium cluster XIVa (such as *Papillibacter cinnamovorans* and *Faecalibacterium prausnitzii*) [41].

Moreover, Ye et al. examined the causal link between GM and biological age acceleration using two-sample Mendelian randomization [42]. They showed that *Streptococcus* is causally associated with bioage acceleration [42].

Persistent low-level inflammation can trigger the development of age-related conditions, including type 2 diabetes (T2DM), obesity, metabolic syndrome, neurodegeneration, cardiovascular disease, and weakened immunity [40].

Dysbiosis impacts physiological functions through various immune mechanisms, contributing to the onset of an inflammatory state. The mechanisms involved encompass disruptions in short-chain fatty acid (SCFA) synthesis, the biosynthesis of amino acids and nucleotides, aberrant catabolism of dietary components, sulfur amino acid metabolism, redox balance, mucin degradation, secretory systems, adhesion, and gene enrichment for invasion [43].

Conversely, in healthy people, GM produces inhibitory metabolites and bacteriocins with antibacterial properties, such as perforating target cells, inhibiting peptidoglycan synthesis, disrupting protein synthesis, and disassembling the DNA of target cells [44]. Bacteriocins boost levels of anti-inflammatory substances and reduce the levels of inflammation-promoting cytokines through various signaling pathways, including mitogen-activated protein kinases (MAPKs) and Toll-like receptors (TLRs) [45]. Specifically, bacteriocin CC34 exhibits anti-inflammatory effects by inhibiting the phosphorylation of inhibitory kappa B kinase beta (IKKβ), nuclear factor of kappa-light-polypeptide-gene-enhancer in B cells inhibitor alpha (IκBα), and nuclear factor kappa-light-chain-enhancer of activated B cells (NF-κB) p65. In lipopolysaccharide (LPS)-treated mice, CC34 significantly decreases the release of inflammatory cytokines TNF-α, IL-1β, and IL-6 in jejunal tissue and serum, while also lowering myeloperoxidase (MPO) levels and reducing pathological damage [45].

The gut barrier is essential for preventing harmful molecules from passing into tissues through the gut mucosa. Research in rodents and non-human primates has demonstrated that gut permeability increases with age, and the expression of tight junction components has been observed to change in several instances [46]. Liu et al. have observed that healthy elderly individuals showed disrupted tight junctions and a reduced expression of E-cadherin and occludin proteins in the colon [47]. In a study comparing them to younger individuals (20–40 years old or 7–12 years old), elderly people (67–77 years old) exhibited a higher expression of claudin-2 and increased intestinal permeability in the ileum, while the expression levels of zonula occludens (ZO)-1, occludin, and JAMA-1 mRNA and protein remained unchanged [48].

GM interacts with the human immune system, enhancing immune cell maturation and function. It accomplishes this through three primary mechanisms: promoting mucin secretion by intestinal goblet cells to uphold the structural integrity of the mucus layer and serve as a barrier; initiating the construction of intestinal mucosa-associated lymphoid tissue; and facilitating the differentiation and maturation of immune cells, primarily through microbiota-induced enhancements that trigger the development of isolated lymphoid follicles (ILFs) for innate defense and activation of naïve T and B cells [49].

For example, Bouskra et al. showed that germ-free mice had underdeveloped intestinal lymphoid tissues, such as crypt nodes and ILFs, as well as reduced levels of serum immunoglobulins [50].

Enteroendocrine cells (EECs) have been observed to express TLRs and activate NF-κB-mediated responses upon exposure to microbial-associated molecular patterns (MAMPs). This reaction results in the secretion of proinflammatory cytokines and enteroendocrine peptides (EEPs) [51]. When exposed to these microbes, there is an observed upregulation in the transcription of genes encoding antimicrobial peptides (AMPs), tachykinin (Tk), and diuretic hormone 31 (DH31). These transcriptional changes are facilitated by IMD pathway signaling in enteroendocrine cells (EECs) that express tachykinin in the anterior midgut [51].

The GM also impacts adaptive immune responses, notably influencing the development and differentiation of CD4^+^ and CD8^+^ T cells. *Lactobacillus* prompts the induction and activation of Tregs, whereas *Clostridium perfringens* G+ increases the proliferation and differentiation of Tregs and Th17 cells, ultimately resulting in IL-17 production by intestinal Th17 cells [52].

Moreover, the GM can influence other immune cells and immunoglobulins by prompting regulatory B cells to generate suppressive cytokines, like IL-10 and TGF-β, effectively dampening inflammation [53].

Furthermore, the GM produces substances that indirectly regulate immune function. SCFAs, which can be metabolized by intestinal epithelial cells and enter the circulatory system, modulate the host immune response by inhibiting histone deacetylase, thus exerting anti-inflammatory effects. For example, butyric acid, a compound specific to certain bacterial strains, acts as an inhibitor of histone deacetylase and as a ligand for G protein-coupled receptors. It serves as a pivotal signaling molecule that influences the host’s immune response [54].

The number of intestinal immune cells does not diminish with age; however, their functionality deteriorates as age increases. For instance, Thevaranjan et al. found that aged macrophages had significantly lower bactericidal capacity compared to younger macrophages [55]. Moreover, aging diminishes the immune capabilities of T cells, primarily by influencing the proliferative reaction of CD4^+^ T cells in Peyer’s patches, lowering the levels of immunoregulatory substances in lamina propria (LP) CD4^+^ T cells, and reducing the occurrence and efficacy of LP CD4^+^ Th17 cells [56]. Wu et al. have reported that systemic inflammation resulted in a heightened expression of the NF-κB transcription factor, activation of the NLRP3 inflammasome, and deficiencies in autophagy signaling pathways, are linked to inflammatory aging [57]. In the elderly, the decline in autophagy activity directly or indirectly triggers the NF-κB pathway, initiating inflammasome activation, and consequently fostering an age-related proinflammatory phenotype [58].

In turn, inflammaging promotes the constant accumulation of recruited immune cells and inflammatory mediators, like TNF-α, IFN-γ, and IL-6 [48]. These elements continuously interact with the intestinal epithelium, as well as with stem cells and Paneth cells in the crypt. This interaction affects the signal transmission and metabolic activity of these cells, which further accelerates the deterioration of the aging intestinal tract and promotes dysbiosis, creating a vicious cycle of inflammaging progression and intestinal aging [59].

Furthermore, there are sex-related differences in the influence of GM on inflammation. For example, Markle et al. showed in a mice model that transferring GM from men to women determined a systemic rise in testosterone and altered their susceptibility to autoimmune diseases [60]. *Akkermansia muciniphila* was less abundant in gonadectomized female mice compared to those with intact gonads. Additionally, the presence of β-estradiol led to an exponential growth in *Akkermansia muciniphila*, demonstrating the bacterium’s responsiveness to female sex hormones [61]. Indeed, the term “estrobolome” has been coined to refer to the collective set of enteric bacterial genes that encode products capable of metabolizing, reactivating, conjugating, and reabsorbing free estrogen [62]. Men had lower α-diversity and a higher presence of *Enterobacteriales* compared to women [35]. In contrast, women exhibited a greater diversity and a higher abundance of *Clostridiales*, suggesting that sex influences the dynamic development of the microbiota [35]. In the elderly, a reduction in sex-related microbial differences in the GM has been observed, likely due to lower levels of sex hormones, especially in postmenopausal women [11].

All the mechanisms mentioned above are summarized in Table 2.

Immunosenescence may also be influenced by the GM. Shen et al., in a study on middle-aged and older adults, showed that microbiome characteristics, such as the richness and abundance of *Akkermansia*, were positively correlated with higher circulating levels of IgG and IgA [65].

In contrast, Kawamoto et al. examined changes in cyclin-dependent kinase inhibitor 2A (p16INK4a) expression in mice [66]. These mice were bred in either a specific pathogen-free (SPF) environment or under germ-free (GF) conditions. The results showed that the age-related increase in p16INK4a expression observed in the abdomens of SPF mice did not occur in the GF environment. Additionally, single-cell transcriptomic and immunohistochemical analyses revealed that ileal germinal center (GC) B cells undergo cellular senescence in a bacteria-dependent manner with aging.

So long as Ileum GC B cells are crucial for IgA production, the cellular senescence of these cells could significantly impact IgA production. These results indicate that, with aging, not only does IgA production decrease, but its diversity also diminishes. The reduction in IgA diversity leads to changes in its binding affinity to gut bacteria, thereby altering the composition of the gut microbiota [67].

Moreover, in the elderly population, the use of multiple medications is often necessary to address a range of conditions. The interaction between different drugs and the GM not only affects the effectiveness of treatment but also influences the structure and function of the microbial ecosystem. Many medications have been demonstrated to undergo partial metabolism by GM, potentially elucidating some of the differences in individual responses, including both efficacy and safety considerations [68].

It is noteworthy to highlight the excessive use of proton pump inhibitors (PPIs). McDonald et al. have linked this practice to a higher susceptibility to intestinal infections, particularly *Clostridium difficile* [69]. Moreover, prolonged use of PPIs has also been associated with alterations in GM in larger-scale investigations [70].

Chronic constipation is a prevalent gastrointestinal condition that impacts both the general populace and the elderly specifically. Its prevalence stands at approximately 20% in individuals over the age of 65 and rises to 30% in those over 84 years old [39]. Specific laxatives have been observed to influence the GM even after short-term usage. Vich Vila et al. have investigated the impact of various medications on the composition and metabolic functions of the GM and highlighted that laxatives showed one of the most pronounced associations with dysbiosis [71].

Numerous studies have confirmed that the GM composition, the function of the gut immune system, and the integrity of the gut barrier play a role in the development of inflammaging in the aging gut. Interestingly, inflammaging also promotes the emergence of aging-like characteristics, such as microbiota imbalance and a weakened intestinal barrier, through a wide range of inflammatory mediators.

## 4. Environmental Factors and Inflammaging

### 4.1. Nutrition and Metabolism

The nutritional status of elderly patients—hospitalized or not—and the role of dietary support as a non-pharmaceutical therapeutic intervention in many diseases, have been considered. Typical features of age-related diseases are cellular stress, metabolic imbalance, oxidative stress, and neuroinflammation [72]. More generally, age-related morbidities are associated with chronic inflammation, notably low-grade inflammation [73].

Against this backdrop, the connection between dietary patterns and systemic markers of inflammation, among them C-reactive protein (RCP), IL-6, and TNF-α, has been explored [74].

Dietary regimes are increasingly considered modulators of chronic inflammation, capable of shaping the GM in a targeted way [75].

The adaptive antitumor immune response is indeed affected by a pro- or anti-inflammatory diet. For instance, diet-based gut inflammation may contribute to colorectal carcinogenesis [73].

Cardiovascular disease (associated with high inflammatory levels) [76], osteoporosis, and sarcopenia were found to be associated with a high intake of dietary advanced glycation end-products (dAGEs) [77]. Moreover, a strong link was noticed between diet-related inflammation and hypertension [78].

Low-grade inflammation is present in obesity and metabolic dysfunction-associated steatotic liver disease (MASLD), involving a high risk of developing fibrosis [79].

High inflammatory scores are correlated with a higher risk of developing all-cause dementia [80].

In this framework, different diets have been compared in terms of their anti- or proinflammatory effects or their impact on autoimmune diseases, infections, and allergies [73]. Specific indices have been used or developed in different studies to measure the impact of dietary regimes on chronic inflammation over time [73,81,82].

A poor nutritional status in elderly patients is often associated with multiple morbidities and polypharmacy. On this basis, some medical associations recommend the implementation of a mandatory assessment of nutritional status for these subjects [83].

Emphasis has been placed on the connection between dietary patterns and systemic markers of inflammation, also when at low grade. A relationship was found between proinflammatory dietary patterns and accelerated aging, which are in turn associated with a higher risk of inflammation-related diseases and the risk of all-cause mortality [84]. In this context, the concept of functional vs. chronological age may be a useful indicator of healthy aging [81].

Both rare and common inflammatory conditions can be affected by dietary patterns (through the GM): diet quality is associated with the health of the GM, notably with microbial genera consistent with pathways related to inflammation and SCFA production [85].

The Mediterranean diet (MD) is undoubtedly the most widely studied dietary pattern. Being rich in unrefined cereals, vegetables and fruit, macro- and micronutrients, omega-3 polyunsaturated fatty acids (PUFA), monounsaturated fatty acids (MUFA), flavonoids, vitamin C and E, the MD can reduce systemic inflammation [74].

The beneficial anti-inflammatory effects of the MD were examined from different perspectives, such as reduced levels of proinflammatory markers, notably, CRP, improved GM composition, subtle gene modulation, and decreased postoperative fecal calprotectin levels [75]. Studies have shown that the MD is associated with longer life expectancy, better mobility in older adults, and a reduced incidence of chronic degenerative diseases and various types of cancer. Additionally, the MD provides several other health benefits, including protecting DNA through antioxidants that neutralize free radicals, promoting intestinal regularity due to higher fiber intake, and enhancing the composition of GM. The PREDIMED study also demonstrated that the MD reduces cardiovascular risk, contrasting with the increased risk linked to the Western dietary pattern [86]. Lower serum concentrations of CRP, IL-6, and TNF-α were identified among patients adhering to MD compared with omnivores [74]. A low advanced glycation end-products diet was associated with lower levels of inflammation markers, such as CRP, IL-6, and TNF-α [77].

Several studies identified a positive role of omega-3 PUFA; recent epidemiological data indicate that the ratio of omega-6/omega-3 PUFA in the diet can influence inflammatory processes in the gastrointestinal tract [75]. More specifically, supplementation of omega-3 PUFA significantly decreased inducible nitric oxide synthase (iNOS) and cyclooxygenase-2 (COX-2) expression [75].

The mitochondrial function appears to be positively modulated by the MD: thanks to polyphenols, fewer free radicals, and mtROS are produced; whole cereals are beneficial for mitochondrial respiration; anti-inflammatory components contribute to high levels of mtDNA content and complex IV activity while modulating mitochondrial enzymes [87].

The MD offers long-term protection against cardiovascular diseases, part of which is mediated by inflammation, uricemia, and renal function. An increased Mediterranean Diet Score over time was found to be associated with lower inflammatory levels and CVD risk [88].

The MD further appears to improve the condition of MASLD patients in terms of body mass index, liver enzymes, and pro-oxidant and proinflammatory status [89]. Adherence to MD is one of the criteria applied to assess the likelihood of developing age-related morbidities.

The ketogenic diet (KD) has been examined in connection with the mitochondrial ketogenic pathway (MKP), by which fatty acids are metabolized into ketones, such as BHB (β-hydroxybutyrate). The latter has been described as having a modulatory effect on systemic and brain inflammation, notably by mitigating the proinflammatory mitochondrial metabolism in microglia. In this connection, BHB has been shown to ameliorate mitochondrial function, rectify the MKP, reduce inflammasome activation, and restore phagocytotic function [90].

There is no consensus about KD’s effect on inflammatory markers, such as CRP, TNF-α, IL-6, IL-8, and IL-10, although some results indicate an improvement in patients over 50 [91].

In this context, a very low-calorie ketogenic diet (VLCKD) can help address obesity and MASLD, both of which are linked to low-grade inflammation and frequently occur together. When implemented in patients with obesity and MASLD, VLCKD seems to cause a simultaneous decrease in white blood cells, platelets, inflammation, liver steatosis, and fibrosis. Therefore, VLCKD is believed to reduce low-grade liver inflammation [79].

Older patients with multiple comorbidities and a health condition that could benefit from a ketogenic diet can experience positive outcomes, including weight loss, improved glucose control, and potentially even some antineoplastic effects. A percentage of older patients (15%) experienced a diet-related adverse event, such as dyslipidemia, constipation, pancreatitis, diarrhea, and fatigue [92].

In contrast, the Western diet (WD) can exert proinflammatory effects. The WD, rich in omega-6 PUFA, saturated fatty acids, high glycemic index carbohydrates, red meat, processed meats, and sugar-sweetened beverages, has been shown to increase serum levels of proinflammatory cytokines [73]. This diet seems to be associated with a higher risk of developing inflammatory gut diseases [75]. In addition to its association with obesity, diabetes, and cardiovascular disease, it induces systemic inflammation [72].

There is a broad consensus that a high intake of saturated fatty acids and/or refined carbohydrates is associated with neuroinflammation, cognitive dysfunction, and neurodegenerative disease [93]. In this way, the WD has been demonstrated to reduce synaptic activity in aged rats [93]. Moreover, the WD supports harmful bacteria that weaken the immune system, producing proinflammatory effects [94].

The impact of the WD on MASLD patients has also been examined: those patients are characterized by higher anthropometric parameters, cardio-metabolic indices, and a lower adherence to the MD than subjects without MASLD [95].

Conway et al. conducted a cross-sectional study, recruiting healthy young participants (aged 18–37 years) and healthy older participants (aged 60 years and above). Participants provided information about their health, diet, and lifestyle, and the levels of CRP, IL6, TNF-α, IL10, IL4, IFN-γ, and granulocyte-macrophage colony-stimulating factor (GM-CSF) were measured [96]. They observed a positive association between WD and high levels of proinflammatory cytokine and Th17 cells.

A dietary pattern characterized by caloric restriction is typical in the elderly and leads to sarcopenia. According to the Working Group on Sarcopenia (EWGSOP2), sarcopenia is a progressive and widespread disorder of skeletal muscle, typically occurring with age progression, and is linked to a higher probability of various adverse outcomes, such as reduced mobility, heightened morbidity, and increased mortality [97].

Morawin et al. aimed to conduct a study to explore the connection between inflammation and sarcopenia, as well as assess the impact of lifestyle on the inflammatory profile [98]. The study showed that CRP, CRP/albumin ratio, IL-1β circulating free DNA (cfDNA), and bilirubin levels were significantly higher in the sarcopenic group compared to the no sarcopenic group [98].

Moreover, several significant metabolic and energy production differences between the sexes could potentially contribute to variations in longevity between men and women.

Men usually exhibit a higher metabolic rate. This difference may also lead to increased production of ROS, oxidative stress, and cellular damage [99]. Additionally, women utilize fat oxidation for total energy expenditure compared to men, particularly in situations of increased energy demand, such as during physical activity [100]. Moreover, women exhibit greater activity of mitochondrial enzymes involved in bioenergetics. This heightened activity could potentially result in increased production of ROS [99].

The level of glutathione is another aging biomarker that may decline with age [101]. Women demonstrate elevated levels of mitochondrial glutathione and other antioxidant enzymes. This difference may provide a protective effect, potentially elucidating the observed variations in longevity between the two sexes [99].

### 4.2. Pollutants

Environmental pollutants can accelerate inflammaging and contribute to age-related diseases [102], including neurodegenerative diseases, chronic obstructive pulmonary disease, and rheumatoid arthritis [103]. For example, Carballo-Carbajal et al. suggested that high levels of the pigment neuromelanin in neurons, associated with exposure to environmental toxicants (e.g., pesticides and heavy metals) and oxidative stress, encourage neurodegeneration and the development of Parkinson’s disease [104].

Sexual dimorphism in the development of pollution-related disorders can be identified. For example, the incidence of several cancers, including mesothelioma and nasopharyngeal cancer, is generally higher in men due to work exposure (e.g., asbestos and wood dust). On the other side, women develop more frequent diseases related to indoor pollutants, for instance, those produced by smoking and heating [105].

Multiple independent studies have associated air pollutants and persistent organic pollutants with an increase in proinflammatory cytokines [106]. Yuan et al. showed elevated plasma levels of IL-6 and IL-10 in a cohort comprising adults residing near an electronic waste site compared to the control group [106]. Exposure to black carbon and particulate matter was linked to alterations in the methylation of IL-6 and IFN-γ as well as hypomethylation of long interspersed nuclear element 1 (LINE-1) in blood cells [107].

In a multicenter study, Pan et al. found that exposure to fine particulate matter (PM_2.5_) accelerates aging in both men and women. Additionally, the authors suggested a protective role of sex hormones against PM_2.5_-related aging [108].

Endocrine disruptors, such as per- and polyfluoroalkyl substances (PFASs), have demonstrated an impact on human physiology by modulating oxidative stress and telomerase activity. As observed by Chaney and Wiley, PFAS exposure has an inverse association with several measures of biological aging, regardless of sex [109].

Exposure to ultraviolet (UV) radiation, smoke, and environmental pollutants induces skin aging, with changes in the skin microbiome (e.g., a reduction in Propionibacterium abundance and an increase in pathogenic Bacteroidetes and Proteobacteria) and physiology, including resident immune cell homeostasis and sebum production [110].

## 5. Therapy Goals and Future Perspectives

With an increasing lifespan, the phenomenon of inflammaging is expected to increase substantially over the next few years.

Age-related alterations in the GM, inflammaging, and immunosenescence can potentially be targeted to maintain immunity and health as we age.

The first goal to be achieved is represented by immunomodulatory interventions. Sauce et al. showed that there is a high rate of early-onset immunosenescence among young adults who had a thymectomy at a very young age [111]. Future studies on thymic regeneration through stem cell therapies and the administration of growth hormone, and their potential to reverse immunosenescence, are still in the initial phases but could greatly benefit future research [17]. “Senolytic drugs” could be used as pharmacological approaches to enhance immune responses in the geriatric population. For example, dasatinib, a tyrosine kinase inhibitor, and quercetin, a flavonoid drug, can be utilized for the targeted elimination of senescent adipocyte progenitors and endothelial cells [112]. Scurr et al. demonstrated that low doses of cyclophosphamide can reduce regulatory T cells in colorectal cancer; similar interventions can be applied to enhance immune responses in the elderly [113].

Another intriguing approach involves the use of stem cells, which in the elderly experience a gradual decline in number, leading to an increased risk of leukemogenesis [114]. However, stem cell therapies targeting this, such as hematopoietic stem cell transplantation in animal models, have shown limited success [17].

In terms of the GM, prebiotics, probiotics, postbiotics, and synbiotics can enhance immunity by supporting intestinal barrier integrity or regulating inflammatory processes. Yang et al. showed that polyphenolic compounds increase the expression of intestinal tight junction proteins and promote the growth of SCFA-producing bacteria [115]. Consuming a polyphenol-rich diet reduces levels of inflammaging in the aging gut and improves gut barrier function [116]. Resveratrol activates sirtuin 1, a histone deacetylase, and reduces inflammatory responses by inhibiting NF-κB and activator protein-1 (AP-1)-dependent signaling pathways [38]. As mentioned earlier, nutrition plays a fundamental role in the composition and function of the GM. The Mediterranean diet and a diet rich in fiber cause an increase in beneficial bacteria, such as *Bifidobacterium* genus, *Alistipes shahii*, *Anaerostipes hadrus*, and *Parabacteroides distasonis*, which can help prevent and counteract aging-related inflammaging [117].

Ren et al. observed that in older mice significant changes in the composition and metabolic function of GM with resistant starch (RS) in their diets, with increased abundance of beneficial bacteria, such as *Bifidobacterium*, *Lactobacillus*, *Roseburia*, and *Faecalibacterium prausnitzii*, resulted in more SCFA and promoted the downregulation of the aging-related gene p53 and the upregulation of the anti-aging gene Sirtuin 1 [118].

Moreover, administration of GOS significantly increased the number of beneficial intestinal bacteria, including species of *Bacteroides* and *Lactobacillus*, in elderly people [38]. It also promoted the activity of intestinal innate immune cells, such as phagocytes and natural killer cells, and increased the expression of the anti-inflammatory factor IL-10, while significantly reducing the production of IL-6, IL-1β, and TNF-α [38].

The progress in fecal microbiota transplantation (FMT) should not be overlooked. In a review, Wang et al. analyzed 782 studies on humans and showed the effectiveness of clinical FMT applications in 85 specific various diseases, grouped into 8 categories: infections, gut diseases, the microbiota–gut–liver axis, the microbiota–gut–brain axis, metabolic diseases, oncology, hematological diseases, and other diseases [119]. The objective of this review was to edit an extensive encyclopedia of clinical FMT reports, drawing attention to the promising potential of FMT for treating dysbiosis-related diseases within the gut and beyond.

Equally important is the role of maintaining an active lifestyle. The post-exercise skeletal muscle inflammatory profile of lifelong-exercising older men resembled that of younger men. In contrast, non-exercising older adults exhibited an elevated expression of TNF-α, TGF-β, and IL-8; this protective effect of lifelong exercise on the inflammatory response was not observed in women [120].

Finally, legislators must also be made aware of the issue because factors, such as pollution and radiation, can exacerbate the effects of immunosenescence and inflammaging through the accumulation of dysfunctional macromolecules and host-derived cell debris of endogenous origin, due to their increased production and impaired or inadequate elimination [17].

## 6. Conclusions

Several lifestyle and pharmacological interventions need to be validated to prevent inflammaging. Furthermore, it would be desirable to validate the use of serum tests, such as CRP, IL-6, leucocyte, and lymphocyte counts, to monitor chronic inflammation in the elderly [121].

Therefore, studies would be desirable to assess the correlation between inflammatory markers and frailty separately in both men and women to discriminate potential disparities in frailty markers among individuals of different sexes, hence contributing to a deeper understanding of the mechanisms underlying frailty onset and progression.

## Figures and Tables

**Figure 1 biomedicines-12-01716-f001:**
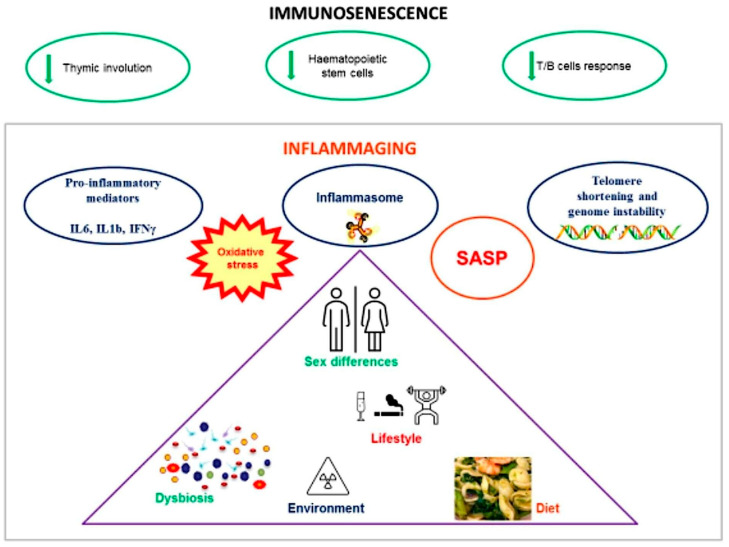
Immunosenescence, characterized by thymic involution, a reduction in hematopoietic stem cells, and decreased humoral immune response, leads to inflammaging. This condition results in the production of proinflammatory cytokines, oxidative stress, DNA instability, and the senescence-associated secretory phenotype (SASP). In turn, inflammaging is influenced by sex, environmental factors, and the dysbiosis of the gut microbiota (GM), which is characterized by increased levels of lipopolysaccharide (LPS), increased mucosa permeability, decreased gut cell autophagy, and an alteration in bile acid metabolism.

**Table 1 biomedicines-12-01716-t001:** Molecular and cellular mechanisms of immunosenescence and inflammaging.

Mechanisms of Immunosenescence and Inflammaging	References
Involution of the thymus	[7]
Dysfunction of hematopoietic stem cells	[7]
Development of SASP by senescent immune cells	[7]
Production of proinflammatory mediators	[8,9]
Decline in T and B cell responses	[15]
Metabolic changes (increased glycolysis, mitochondrial dysfunction, ROS)	[7,18]
Telomere shortening and genome instability	[21,22]
Immune phenotypic changes in senescent immune cells	[26,27]

Abbreviations: SASP, senescence-associated secretory phenotype; ROS, reactive oxygen species.

**Table 2 biomedicines-12-01716-t002:** Age-related changes in GM.

Mechanism	Elderly GM	Effect	References
Lipopolysaccharides	Higher levels	Promotes TNF-α, IFN-β, IL-12, IL-1β, and IL-6	[37]
Bile acid metabolism	Decrease in the bacterial enzyme bile salt hydrolase (BSH), expressed by Firmicutes, Bacteroidetes, and Actinobacteria	Failure to activate genes involved in intestinal conservation, suppression of bacterial proliferation, and maintenance of mucosal barrier integrity	[63]
Dysbiosis	Reduction in Bifidobacteria, Firmicutes, Clostridium cluster IV, and Clostridium cluster XIVa	Inhibits the IL-6/Stat3/IL-17 pathway, reducing the differentiation of CD4^+^ T cells into Th17 cells	[41,64]
Gut permeability	Increased	Disrupted tight junctions and reduced expression of E-cadherin and occludin proteins	[47]
Intestinal immune cells	Functionality deteriorates	Lowers the levels of immunoregulatory substances in LP CD4^+^ T cells and reduces the occurrence and efficacy of LP CD4^+^ Th17 cells	[56]
Gut cell autophagy	Reduced	Triggers the NF-κB pathway, initiating inflammasome activation	[58]

Abbreviations: GM, gut microbiota; TNF-α, tumor necrosis factor-alpha; IFN-β, interferon beta; IL, interleukin; Th, T helper; LP, lamina propria; NF-κB, nuclear factor kappa-light-chain-enhancer of activated B cells.

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
