# Peer review of "Inflammaging: The Next Challenge—Exploring the Role of Gut Microbiota, Environmental Factors, and Sex Differences"

_biomedicines, 2024, doi:10.3390/biomedicines12081716_

Round 1

Reviewer 1 Report

Comments and Suggestions for Authors

The paper addresses an important and timely topic on chronic inflammation, microbiota, frailty, and other factors.

Comments as below:

1. It is not clearly described the details on the literature search strategy, inclusion/exclusion criteria, and data analysis methods would strengthen the paper. It's unclear whether this is a systematic review, meta-analysis, or narrative review.

3. The authors discussed many potential relation from lab. However, the most important evidence supporting the microbiota and inflammation related diseases is FMT in practice. I strongly suggest to read the review: Encyclopedia of fecal microbiota transplantation: a review of effectiveness in the treatment of 85 diseases. Chin Med J. 2022;135(16):1927-1939. The scope would be understood fully from these evidences.

4. There are some minor grammatical errors that should be addressed (e.g., in line 557: "Therefore, would be desirable studies" should be "Therefore, studies would be desirable").

Comments on the Quality of English Language

The language is fine.

Author Response

Dear Editor of Biomedicines

First, my coauthors and I would like to thank you sincerely for this opportunity to cooperate. We profoundly thank the reviewers for the comments and useful suggestions to improve the paper. We thank You for your constructive critique and hope the review process has improved the manuscript. If additional changes are warranted, we will make them. 

We hope that this revised version of our manuscript may now be found suitable for publication. 

This is a point-by-point list of changes made in the paper:

REVIEWER 1

  1. It is not clearly described the details on the literature search strategy, inclusion/exclusion criteria, and data analysis methods would strengthen the paper. It's unclear whether this is a systematic review, meta-analysis, or narrative review.

We have clarified the criteria for the literature search and specified that this is a ‘narrative review’.

  1. The authors discussed many potential relation from lab. However, the most important evidence supporting the microbiota and inflammation related diseases is FMT in practice. I strongly suggest to read the review: Encyclopedia of fecal microbiota transplantation: a review of effectiveness in the treatment of 85 diseases. Chin Med J. 2022;135(16):1927-1939. The scope would be understood fully from these evidences.

We have included the reference and discussed it, as requested.

  1. There are some minor grammatical errors that should be addressed (e.g., in line 557: "Therefore, would be desirable studies" should be "Therefore, studies would be desirable").

We have corrected the grammatical errors, as requested.

We thank You for your constructive critique and hope the review process has improved the manuscript.

If additional changes are warranted, we will make them.

We hope that this revised version of our manuscript may now be found suitable for publication.

Sincerely,

Rossella Cianci

Reviewer 2 Report

Comments and Suggestions for Authors

1)    The present review was performed between 2019 and 2024, namely just 5 years!   It seems to be too short term as so called systematic reviewing".   If they have an evident reason for that, they must show in the introduction (around the 69th line or somewhere).

2)    On the other hand, the final conclusion part was too long!  They have to shorten that in order to be clear the goal.  

period!  between 2019 and 2024

Author Response

Dear Editor of Biomedicines

First, my coauthors and I would like to thank you sincerely for this opportunity to cooperate. We profoundly thank the reviewers for the comments and useful suggestions to improve the paper. We thank You for your constructive critique and hope the review process has improved the manuscript. If additional changes are warranted, we will make them. 

We hope that this revised version of our manuscript may now be found suitable for publication. 

This is a point-by-point list of changes made in the paper:

REVIEWER 2

1)    The present review was performed between 2019 and 2024, namely just 5 years!   It seems to be too short term as so called systematic reviewing".   If they have an evident reason for that, they must show in the introduction (around the 69th line or somewhere).

We have modified the text, because there was a typo (2014-2024) and we have written ‘in the last ten years’.

2)    On the other hand, the final conclusion part was too long!  They have to shorten that in order to be clear the goal.

We have separated the targets and therapeutic perspectives from the conclusions to make the reading more easy.

We thank You for your constructive critique and hope the review process has improved the manuscript.

If additional changes are warranted, we will make them.

We hope that this revised version of our manuscript may now be found suitable for publication.

Sincerely,

Rossella Cianci

Reviewer 3 Report

Comments and Suggestions for Authors

The authors are presenting their work regarding review and summarisation of impact on gut microbiota on health and ageing. The article is relevant and very well presented. I only have a few minor changes that I recommend before the manuscript can be accepted for publication.

Throughout the manuscript,

A. Correct the line breaks so that they do not occur in the middle of words.

B. I recommend not to write that inflammaging will be a pandemic. Inflammaging is defined as a low-grade chronic, sterile, inflammation state in the body that often occur as we age. It is not a disease by itself but can result in disease. A pandemic is defined as an epidemic spread over several countries or continents and that affects a large number of people. Due to this I advice not to call inflammaging a pandemic, as done in the title and summary.  

Specific corrections that I recommend

1)       Row 35, I suggest to write “… progression of disequilibrium…” or “… advancement of disequilibrium…” instead of “… the worsening disequilibrium…”.

2)       Row 35-37, Add reference to this sentence.

3)       Row 59, I suggest to write “in the complex crosstalk” instead of “in complex crosstalk”.

4)       Row 60, I suggest to write “… has been evaluated…” instead of “… could be evaluated…”.

5)       Row 79-81, I suggest to write “…adults found that increased levels of the pro-inflammatory C-XC motif chemokine ligand 10 (CXCL10), IL-2, IL-4, IL-6, IL-17, TNF-α, and C-C motif chemokine ligand 11 (CCL11) were positively associated with progression of age…” instead of “…adults found that the pro-inflammatory C-X-C motif chemokine ligand 10 (CXCL10), IL-2, IL-4, IL-6, IL-17, TNF-α, and C-C motif chemokine ligand 11 (CCL11) have a positive association with age…”.

6)       Row 93, I suggest to write “…independent of gender…” instead of “… of both sexes…”

7)       Row 96, I suggest to write “…which have immunosuppressive roles [13].” Instead of “… with an immunosuppressive role [13].”.

8)       Row 105, I suggest to omit “repertoire”.

9)       “…described the reduced function of macrophages, monocytes, neutrophils, and NK cells, the impairment in T cell activation by dendritic cells, and changes in adaptive immunity, such as the reduced output…Row 126, I suggest to write “…increased triglyceride synthesis,…” instead of “… higher triglyceride synthesis,…”

10)    Row 130, I suggest to write “… as compared to women…” instead of “… compared to women…”

11)    Row 133-135, I suggest to write “However, after menopause depletion of estrogens, which exert antioxidant functions, increases oxidative damage in women causing less pronounced differences between the genders[19].” instead of “However, after menopause, the depletion of estrogens, which exert antioxidant functions, increases oxidative damage in women, resulting in less pronounced sex differences [19].”

12)    Row 137-139, Add reference to this sentence.

13)    Row 167-169, Add reference to this sentence.

14)    Row 172, I suggest to write “…as illustrated in figure 1 [27].” instead of “… [27]. Figure 1”

15)    Row 189-191, I suggest to write “…described reduced function of macrophages, monocytes, neutrophils, and NK cells, impairment in T cell activation by dendritic cells, and changes in adaptive immunity, such as reduced output…” instead of “…described the reduced function of macrophages, monocytes, neutrophils, and NK cells, the impairment in T cell activation by dendritic cells, and changes in adaptive immunity, such as the reduced output…”.

16)    “…described the reduced function of macrophages, monocytes, neutrophils, and NK cells, the impairment in T cell activation by dendritic cells, and changes in adaptive immunity, such as the reduced output…

17)    “…described the reduced function of macrophages, monocytes, neutrophils, and NK cells, the impairment in T cell activation by dendritic cells, and changes in adaptive immunity, such as the reduced output…

18)    Row 198, I suggest to write “…broad spectrum of…” instead of “…vast selection of…”

19)    Row 212, I suggest to write “…elderly populations can have a reduced…” instead of “…that the elderly have a reduced…”

20)    Row 217-218, I suggest to write “Moreover, Ye et al. examined the causal link between GM and biological age acceleration using two-sample Mendelian randomization [37].” instead of “Moreover, Ye et al. in a two-sample Mendelian randomization examine the causal link between GM and biological age acceleration [37].”

21)    Row 501-502, I suggest to write “…With an increasing lifespan, the phenomenon of inflammaging is expected to increase substantially over the next years” instead of “…Due to the increase in the average age of the population, the phenomenon of inflammaging is shaping up to be a new pandemic in the next years. ”

22)    Row 504, I suggest to write “…as we age” instead of “…during the elderly.” 

Comments on the Quality of English Language

The authors are presenting their work regarding review and summarisation of impact on gut microbiota on health and ageing. The article is relevant and very well presented. I only have a few minor changes that I recommend before the manuscript can be accepted for publication.

Throughout the manuscript,

A. Correct the line breaks so that they do not occur in the middle of words.

B. I recommend not to write that inflammaging will be a pandemic. Inflammaging is defined as a low-grade chronic, sterile, inflammation state in the body that often occur as we age. It is not a disease by itself but can result in disease. A pandemic is defined as an epidemic spread over several countries or continents and that affects a large number of people. Due to this I advice not to call inflammaging a pandemic, as done in the title and summary.  

Specific corrections that I recommend

1)       Row 35, I suggest to write “… progression of disequilibrium…” or “… advancement of disequilibrium…” instead of “… the worsening disequilibrium…”.

2)       Row 35-37, Add reference to this sentence.

3)       Row 59, I suggest to write “in the complex crosstalk” instead of “in complex crosstalk”.

4)       Row 60, I suggest to write “… has been evaluated…” instead of “… could be evaluated…”.

5)       Row 79-81, I suggest to write “…adults found that increased levels of the pro-inflammatory C-XC motif chemokine ligand 10 (CXCL10), IL-2, IL-4, IL-6, IL-17, TNF-α, and C-C motif chemokine ligand 11 (CCL11) were positively associated with progression of age…” instead of “…adults found that the pro-inflammatory C-X-C motif chemokine ligand 10 (CXCL10), IL-2, IL-4, IL-6, IL-17, TNF-α, and C-C motif chemokine ligand 11 (CCL11) have a positive association with age…”.

6)       Row 93, I suggest to write “…independent of gender…” instead of “… of both sexes…”

7)       Row 96, I suggest to write “…which have immunosuppressive roles [13].” Instead of “… with an immunosuppressive role [13].”.

8)       Row 105, I suggest to omit “repertoire”.

9)       “…described the reduced function of macrophages, monocytes, neutrophils, and NK cells, the impairment in T cell activation by dendritic cells, and changes in adaptive immunity, such as the reduced output…Row 126, I suggest to write “…increased triglyceride synthesis,…” instead of “… higher triglyceride synthesis,…”

10)    Row 130, I suggest to write “… as compared to women…” instead of “… compared to women…”

11)    Row 133-135, I suggest to write “However, after menopause depletion of estrogens, which exert antioxidant functions, increases oxidative damage in women causing less pronounced differences between the genders[19].” instead of “However, after menopause, the depletion of estrogens, which exert antioxidant functions, increases oxidative damage in women, resulting in less pronounced sex differences [19].”

12)    Row 137-139, Add reference to this sentence.

13)    Row 167-169, Add reference to this sentence.

14)    Row 172, I suggest to write “…as illustrated in figure 1 [27].” instead of “… [27]. Figure 1”

15)    Row 189-191, I suggest to write “…described reduced function of macrophages, monocytes, neutrophils, and NK cells, impairment in T cell activation by dendritic cells, and changes in adaptive immunity, such as reduced output…” instead of “…described the reduced function of macrophages, monocytes, neutrophils, and NK cells, the impairment in T cell activation by dendritic cells, and changes in adaptive immunity, such as the reduced output…”.

16)    “…described the reduced function of macrophages, monocytes, neutrophils, and NK cells, the impairment in T cell activation by dendritic cells, and changes in adaptive immunity, such as the reduced output…

17)    “…described the reduced function of macrophages, monocytes, neutrophils, and NK cells, the impairment in T cell activation by dendritic cells, and changes in adaptive immunity, such as the reduced output…

18)    Row 198, I suggest to write “…broad spectrum of…” instead of “…vast selection of…”

19)    Row 212, I suggest to write “…elderly populations can have a reduced…” instead of “…that the elderly have a reduced…”

20)    Row 217-218, I suggest to write “Moreover, Ye et al. examined the causal link between GM and biological age acceleration using two-sample Mendelian randomization [37].” instead of “Moreover, Ye et al. in a two-sample Mendelian randomization examine the causal link between GM and biological age acceleration [37].”

21)    Row 501-502, I suggest to write “…With an increasing lifespan, the phenomenon of inflammaging is expected to increase substantially over the next years” instead of “…Due to the increase in the average age of the population, the phenomenon of inflammaging is shaping up to be a new pandemic in the next years. ”

22)    Row 504, I suggest to write “…as we age” instead of “…during the elderly.”

Author Response

Dear Editor of Biomedicines

First, my coauthors and I would like to thank you sincerely for this opportunity to cooperate. We profoundly thank the reviewers for the comments and useful suggestions to improve the paper. We thank You for your constructive critique and hope the review process has improved the manuscript. If additional changes are warranted, we will make them. 

We hope that this revised version of our manuscript may now be found suitable for publication. 

This is a point-by-point list of changes made in the paper:

REVIEWER 3

The authors are presenting their work regarding review and summarisation of impact on gut microbiota on health and ageing. The article is relevant and very well presented. I only have a few minor changes that I recommend before the manuscript can be accepted for publication. 

Throughout the manuscript,

  1. Correct the line breaks so that they do not occur in the middle of words. 
  2. I recommend not to write that inflammaging will be a pandemic. Inflammaging is defined as a low-grade chronic, sterile, inflammation state in the body that often occur as we age. It is not a disease by itself but can result in disease. A pandemic is defined as an epidemic spread over several countries or continents and that affects a large number of people. Due to this I advice not to call inflammaging a pandemic, as done in the title and summary.  

Specific corrections that I recommend

1)       Row 35, I suggest to write “… progression of disequilibrium…” or “… advancement of disequilibrium…” instead of “… the worsening disequilibrium…”.

2)       Row 35-37, Add reference to this sentence.

3)       Row 59, I suggest to write “in the complex crosstalk” instead of “in complex crosstalk”.

4)       Row 60, I suggest to write “… has been evaluated…” instead of “… could be evaluated…”.

5)       Row 79-81, I suggest to write “…adults found that increased levels of the pro-inflammatory C-XC motif chemokine ligand 10 (CXCL10), IL-2, IL-4, IL-6, IL-17, TNF-α, and C-C motif chemokine ligand 11 (CCL11) were positively associated with progression of age…” instead of “…adults found that the pro-inflammatory C-X-C motif chemokine ligand 10 (CXCL10), IL-2, IL-4, IL-6, IL-17, TNF-α, and C-C motif chemokine ligand 11 (CCL11) have a positive association with age…”.

6)       Row 93, I suggest to write “…independent of gender…” instead of “… of both sexes…”

7)       Row 96, I suggest to write “…which have immunosuppressive roles [13].” Instead of “… with an immunosuppressive role [13].”.

8)       Row 105, I suggest to omit “repertoire”.

9)       “…described the reduced function of macrophages, monocytes, neutrophils, and NK cells, the impairment in T cell activation by dendritic cells, and changes in adaptive immunity, such as the reduced output…Row 126, I suggest to write “…increased triglyceride synthesis,…” instead of “… higher triglyceride synthesis,…”

10)    Row 130, I suggest to write “… as compared to women…” instead of “… compared to women…”

11)    Row 133-135, I suggest to write “However, after menopause depletion of estrogens, which exert antioxidant functions, increases oxidative damage in women causing less pronounced differences between the genders[19].” instead of “However, after menopause, the depletion of estrogens, which exert antioxidant functions, increases oxidative damage in women, resulting in less pronounced sex differences [19].”

12)    Row 137-139, Add reference to this sentence.

13)    Row 167-169, Add reference to this sentence.

14)    Row 172, I suggest to write “…as illustrated in figure 1 [27].” instead of “… [27]. Figure 1”

15)    Row 189-191, I suggest to write “…described reduced function of macrophages, monocytes, neutrophils, and NK cells, impairment in T cell activation by dendritic cells, and changes in adaptive immunity, such as reduced output…” instead of “…described the reduced function of macrophages, monocytes, neutrophils, and NK cells, the impairment in T cell activation by dendritic cells, and changes in adaptive immunity, such as the reduced output…”.

16)    “…described the reduced function of macrophages, monocytes, neutrophils, and NK cells, the impairment in T cell activation by dendritic cells, and changes in adaptive immunity, such as the reduced output…

17)    “…described the reduced function of macrophages, monocytes, neutrophils, and NK cells, the impairment in T cell activation by dendritic cells, and changes in adaptive immunity, such as the reduced output…

18)    Row 198, I suggest to write “…broad spectrum of…” instead of “…vast selection of…”

19)    Row 212, I suggest to write “…elderly populations can have a reduced…” instead of “…that the elderly have a reduced…”

20)    Row 217-218, I suggest to write “Moreover, Ye et al. examined the causal link between GM and biological age acceleration using two-sample Mendelian randomization [37].” instead of “Moreover, Ye et al. in a two-sample Mendelian randomization examine the causal link between GM and biological age acceleration [37].”

21)    Row 501-502, I suggest to write “…With an increasing lifespan, the phenomenon of inflammaging is expected to increase substantially over the next years” instead of “…Due to the increase in the average age of the population, the phenomenon of inflammaging is shaping up to be a new pandemic in the next years. ”

22)    Row 504, I suggest to write “…as we age” instead of “…during the elderly.” 

Thank you for your feedback. We have addressed and corrected all the points suggested, as requested.

We thank You for your constructive critique and hope the review process has improved the manuscript.

If additional changes are warranted, we will make them.

We hope that this revised version of our manuscript may now be found suitable for publication.

Sincerely,

Rossella Cianci

Reviewer 4 Report

Comments and Suggestions for Authors

This review is very interesting, but major modifications are needed in the key descriptive content of the entire manuscript.

1. In the introduction section, the impact of aging on gut microbiota is introduced (Line 55, Line 62), while the subsequent review Gut Microbiota and Inflammaging focuses on how gut microbiota affects inflammation and leads to inflammaging. Although these two factors interact with each other, the introduction section about gut microbiota should be revised to how gut microbiota lead to inflammaging according to the purpose--“Understanding how immunity, GM, and environmental factors act in elderly individuals may allow the design of tailored therapeutic interventions aimed at promoting healthy aging”.

2. How immunosenescence lead to inflammaging should be expected to be reviewed in the section “Immunosenescence and Inflammaging”, but most of the section focuses on the changes in immune responses and inflammatory mediators during the aging process.

3. The manuscript should focus on how immunosenescence and gut microbiota affect inflammaging, so as to design tailored therapeutic interventions aimed at promoting healthy aging, just like the review on diet and pollutants in the manuscript.

Author Response

Dear Editor of Biomedicines

First, my coauthors and I would like to thank you sincerely for this opportunity to cooperate. We profoundly thank the reviewers for the comments and useful suggestions to improve the paper. We thank You for your constructive critique and hope the review process has improved the manuscript. If additional changes are warranted, we will make them. 

We hope that this revised version of our manuscript may now be found suitable for publication. 

This is a point-by-point list of changes made in the paper:

REVIEWER 4

This review is very interesting, but major modifications are needed in the key descriptive content of the entire manuscript.

  1. In the introduction section, the impact of aging on gut microbiota is introduced(Line 55, Line 62), while the subsequent review “Gut Microbiota and Inflammaging” focuses on how gut microbiota affects inflammation and leads to  Although these two factors interact with each other, the introduction section about gut microbiota should be revised to “how gut microbiota lead to inflammaging” according to the purpose--“Understanding how immunity, GM, and environmental factors act in elderly individuals may allow the design of tailored therapeutic interventions aimed at promoting healthy aging”.

We have modified the text, as suggested.

  1. “Howimmunosenescence lead to inflammaging” should be expected to be reviewed in the section “Immunosenescence and Inflammaging”, but most of the section focuses on the changes in immune responses and inflammatory mediators during the aging process.

We have integrated the latest scientific research on immunosenescence and modified the text as requested.

  1. Themanuscript should focus on how immunosenescence and gut microbiota affect inflammaging, so as to design tailored therapeutic interventions aimed at promoting healthy aging, just like the review on diet and pollutants in the manuscript.
    We have added information on the gut microbiota and immunosenescence. The therapeutic targets are described in detail in the apposite section.

We thank You for your constructive critique and hope the review process has improved the manuscript.

If additional changes are warranted, we will make them.

We hope that this revised version of our manuscript may now be found suitable for publication.

Sincerely,

Rossella Cianci

Round 2

Reviewer 4 Report

Comments and Suggestions for Authors

No comments. The authors answered the questions and revised the entire manuscript.